# Epidemiological Study of Risk Factors for Lung Cancer in KwaZulu-Natal, South Africa

**DOI:** 10.3390/ijerph19116752

**Published:** 2022-05-31

**Authors:** Noluthando P. Mbeje, Themba Ginindza, Nkosana Jafta

**Affiliations:** 1Cancer and Infectious Diseases Epidemiology Research Unit, College of Health Sciences, University of KwaZulu-Natal, Durban 4041, South Africa; ginindza@ukzn.ac.za (T.G.); jaftan@ukzn.ac.za (N.J.); 2Discipline of Public Health Medicine, School of Nursing and Public Health, College of Health Sciences, University of KwaZulu-Natal, Durban 4041, South Africa; 3Discipline of Occupational and Environmental Health, School of Nursing and Public Health, College of Health Sciences, University of KwaZulu–Natal, Durban 4041, South Africa

**Keywords:** risk factors, lung cancer, carcinogens, case-control, KwaZulu-Natal

## Abstract

The high incidence cancer rates are due to factors such as behavior, occupational exposures, genetics, environmental pollution and infections. The aim of this study was to identify risk factors associated with lung cancer among patients seen in the public health facilities in KwaZulu-Natal, South Africa. In this case-control study, 75 cases and 159 controls were interviewed using a structured close-ended questionnaire. Logistic regression showed a positive association between lung cancer and tobacco smoking (OR = 2.86, 95% CI: 1.21–6.77) and exposure to passive smoke (OR = 3.28, 95% CI: 1.48–7.30). When adjusted for other covariates, tobacco smoking and passive smoke were still positively associated with increased risk of lung cancer. Alcohol consumption (aORs ranging from 2.79 to 3.35) and history of lung disease (aORs ranging from 9.91 to 12.1) were statistically significantly associated with lung cancer. Our study suggests that tobacco smoke exposure is the major cause of lung cancer, and increased exposure to occupational and environmental carcinogenic substances, alcohol consumption and history of lung disease increase the risk of lung cancer. Based on our findings, policy development and planning of prevention strategies incorporating smoking legislations, occupational health and safety are essential in South Africa.

## 1. Introduction

Globally, lung cancer is the most common cancer and has one of the poorest prognoses and highest mortality rates [1], and this has resulted in it being the leading cause of cancer death (18.4% of the total cancer deaths) with 1.8 million related lung cancer deaths in 2018 [2]. Cancer morbidity and mortality are associated with factors such as behavior, occupational exposures, genetics, environmental pollution and infections [2,3].

Lung cancer mortality in South Africa is high [4] with the age standardized incidence rate (ASR) 8.37/100,000 for males and 3.57/100,000 for females [5]. Prior studies on lung cancer in South Africa indicate that the lung cancer mortality rate is high [4,6]. Other studies indicate the need for screening and management of lung cancer patients. According to Eeden et al., smoking is the main risk factor for lung cancer; however, data on smoking prevalence need to be updated [6]. Other risk factors associated with lung cancer were identified in this study.

The association between tobacco smoking and lung cancer is well established [7,8], with some studies attributing approximately 80% of global lung cancer mortality to tobacco smoking [7,9]. Secondhand smokers are also at risk for lung cancer [4]. It is estimated that approximately 15% of lung cancer cases are people that are light smokers or never smoked [10]; therefore, there are more risk factors associated with lung cancer other than smoking [11]. Fifteen percent (15%) of global lung cancer mortality is attributed to occupational risk factors such as asbestos and silica [12].

A systematic review and meta-analysis by Ngamwong et al. found that the combination of tobacco smoking and asbestos exposure causes lung cancer through an additive mechanism [13]. Another study found that lung cancer risk in individuals who are tobacco smokers diagnosed with chronic obstructive pulmonary disease (COPD) was more than 11-fold higher compared to individuals that only smoke [14]. A meta-analysis involving 31 studies found a 2-fold increased risk of lung cancer in persons with a family history of lung cancer with most of them being nonsmokers [15].

In Africa, cancer is a cumulatively persisting challenge due to the aging and growth of the population, as well as the increasing prevalence of risk factors [16]. The prevalence of smoking tobacco has increased to 37% for males and significantly dropped to 8% for 57 women aged 15 and older [17]. The South African demographic and health survey indicated that among smokers, most smokers smoke cigarettes on a daily basis at 30% for males and 6% for females [17]. The objective of the study was to identify and quantify risk factors of lung cancer amongst patients seen in the public health facilities in KwaZulu-Natal, South Africa.

## 2. Materials and Methods

### 2.1. Study Setting and Population

This was a case-control designed study with a ratio of 1:2, conducted from January 2018 to March 2019. The participants were recruited from the three public health facilities that provide oncology services in the province of KwaZulu-Natal, South Africa, namely: Addington Hospital, Greys Hospital and Inkosi Albert Luthuli Central Hospital (IALCH).

### 2.2. Identification and Recruitment of Participants

Cases were patients with the lung cancer diagnosis that was laboratory confirmed through a biopsy, X-ray, CT Scan and cytology, and/or clinical investigations. All the details regarding cancer diagnoses, stage, comorbidities and other exposure were self-reported during patient interviews. The cases were recruited at the oncology clinic during the patient’s lung cancer treatment appointment at the three public health facilities. The cases had to be patients that were diagnosed with cancer for the first time (primary cancer), and those that had a previous diagnosis of cancer were excluded.

In the same health facilities, controls were recruited from the outpatient clinics and patient wards of the orthopedics departments. The controls were patients that had not been diagnosed with cancer before and had no signs and symptoms similar to those of lung cancer or respiratory illness at the time of the study. The controls were screened by asking questions on lung cancer symptoms or signs such as unexplained cough, shortness of breath or whizzing, blood sputum, chest pains and difficulty in swallowing.

### 2.3. Data Collection Using Questionnaire

After obtaining consent, the participants were interviewed using a structured close-ended questionnaire made up of questions that inquire about the patient’s demographics, the diagnosis and different exposures and risk factors associated with lung cancer. The information collected on diagnosis included lung cancer symptoms and/or signs (unexplained cough, blood sputum, shortness of breath/whizzing, hoarseness of voice, chest pains, unexplained weight-loss); date and age at diagnosis; name of facility where the cancer was diagnosed; method of cancer diagnoses, cancer treatment and referral process were also collected. Information on other diseases such as high blood pressure, pneumonia, chronic bronchitis, asthma and infectious disease such as HIV/AIDS and tuberculosis was also covered. Family history information on any family member of the participant that had been diagnosed with any type of cancer was also collected. With regards to information on occupational history and exposure to likely carcinogenic substances, we collected information on industry and type of work, duration at the company, how participants were exposed to different substances and use of protective personal equipment. Additional information on personal habits and behaviors such as smoking tobacco (cigarette) and exposure to passive smoke at home and/or the workplace and use of alcohol was also collected (Figure 1). The questionnaires were either administered face-to-face or by telephone in isiZulu or English at a time convenient for the participants.

### 2.4. Data Analysis

All completed questionnaires were captured on the REDCap database (REDCap v8.11.7, Vanderbilt University, TN, USA). Thereafter, data were checked and corrected for possible errors prior to analysis. Of the recruited 79 cases and 160 controls, four cases and one control were excluded from data analysis due to previous participants’ history of cancer diagnosis, therefore leaving 234 participants (75 lung cancer cases and 159 controls) (Figure 2). Data were then transferred from REDCap to Stata IC version 13 (StataCorp, College Station, TX, USA) for further analysis.

The characteristics of lung cancer cases and controls participating in the study, namely, the age, gender, race, education and socioeconomic, were described using descriptive statistics. Passive smoke in this study was defined as exposure to cigarette smoke at home (someone smoking inside the house) and in the workplace. History of lung disease was categorized as positive if the participants had been diagnosed with tuberculosis, asthma, pneumonia or chronic bronchitis. The occupations or industries that the participants were working in or have worked in were grouped according to the Statistics South Africa (Stats-SA) Industry code list. Industry 1—clerical support, services and sales, elementary occupation (Stats-SA code: 61, 62,64, 81–84, 93–99); Industry 2—transport and repair of motor vehicles (Stats-SA code: 63, 71–75), Industry 3—mining, construction, manufacturing (Stats-SA code: 21–50); and Industry 4—skilled agriculture, hunting, forestry/fishing (Stats-SA code:11–13) [18].

The differences in characteristics between cases and controls were tested using simple logistic regression. Associations between the risk factors and lung cancer were assessed using multivariable logistic regression. We ran six models; two models were for environmental exposures, and four models were for occupational exposures. The first models were on smoking and passive smoke. The second set of models we ran was on occupational and carcinogen exposures; these models included smoking as a covariate. The four models on occupational exposure had different independent variables/indicators, namely, (1) office and household; (2) transport and repair of motor vehicles; (3) mining, construction, all types of manufacturing; (4) agriculture, hunting, forestry/fishing as per the classification of Stats-SA. Significant covariates in the univariate model and/or those that were indicated by the literature to be confounders were selected a priori for inclusion in the models. Variables such as age, gender, race, marital status and education were included into all the models regardless of the significance in the univariate analysis.

### 2.5. Ethical Considerations

The study was approved by the University of KwaZulu-Natal’s Biomedical Research Ethics Committee (BREC) (Ref: BE533/18) and the province of the KwaZulu-Natal Department of Health (Ref: HRKM0007/18 (KZ_201B01_013)). Permission to conduct the study was also granted by the three public health facilities at which the lung cancer risk factors questionnaires were administered. All potential participants were informed about the study in their preferred language (in isiZulu or English) and were given an information sheet explaining the study. Consents were obtained from participants who agreed to be part of the study before the questionnaires were administered.

## 3. Results

Demographic characteristics of 234 participants (75 cases and 159 controls) are shown in Table 1. Among the recruited study participants, for both cases and controls, males were of a higher proportion at 64.0% and 60.4%, respectively. The mean (SD) age was higher for the cases at 61.8 (13.9) years than for the controls at 54.8 (14.4) years. The majority of the cases were married (57.5%), whereas the majority of the controls were single (44.5%). Approximately 70% of participants had an educational level ranging between high school to higher education. A higher proportion of the cases (71%) and controls (74%) had a monthly household income of R0 to R4000.

For the lung cancer cases, biopsy was the most commonly used basis of diagnosis at 74.7% (*n* = 56). The stage of lung cancer was unknown for more than half (58.7%, *n* = 44) of the cancer cases. For the cancer cases that were staged, stage 4 was the most prevalent at 37.3% (*n* = 28). According to our data (Table 2), 65.3% (*n* = 49) of the cases were tobacco smokers, and 54.7% (*n* = 41) of the cases reported that they were exposed to passive smoke either at home or in the workplace.

A first-degree relative family history of cancer was crudely associated with increased risk of lung cancer in siblings by 8-fold (unadjusted OR 8.93 (95% CI: 2.41–33.1; *p* = 0.001), but for biological parents this was not the case. History of lung disease (tuberculosis and asthma) was crudely associated with lung cancer by more than 7-fold (unadjusted OR 7.83 (95% CI 3.29 –18.7; *p* < 0.001).

More cases vs. controls were exposed to soot (9.33% vs. 1.26%) and iron and steel (10.7% vs. 1.26%), and there was an unadjusted increased risk of OR = 8.08 (95% CI: 1.64–39.91; *p* = 0.010) and OR = 9.37 (95% CI: 1.94–45.31; *p* = 0.005), respectively. Environmental asbestos exposure, occupational asbestos exposure, radon were not statistically associated with lung cancer but were positively associated at OR = 1.08 (95% CI: 0.58–2.02; *p* = 0.805), OR = 1.53 (95% CI: 0.56–4.20; *p* = 0.405) and OR = 4.33 (95% CI: 0.39–48.5; *p* = 0.235), respectively.

Table 3 contains results of multivariable logistic regression showing a positive association of lung cancer for tobacco smoking and exposure to passive smoke. After adjusting for covariates, tobacco smoking (aOR = 2.86, 95% CI: 1.21–6.77; *p* = 0.017) and passive smoke (aOR = 3.28, 95% CI: 1.48–7.30; *p* = 0.004) were still positively associated with an increased risk among cases when compared to people not exposed to tobacco; alcohol consumption (aORs ranging from 2.79 to 3.35) and history of lung disease (aORs ranging from 9.91 to 12.1) were significantly associated with lung cancer in the multivariable logistic models.

Working in mining, construction and manufacturing was a significant risk factor even after adjusting covariates (aOR = 2.64, 95% CI: 1.16–6.01; *p* = 0.020), although the risk was amplified compared to when it was not adjusted (OR = 2.15, 95% CI: 1.22–3.79). Increased unadjusted risk of lung cancer was observed in participants with a history of working in transport, storage, repair of motor vehicles (OR = 3.05, 95% CI: 1.38–6.72; *p* = 0.006) and after adjusting covariates (aOR = 1.63, 95% CI: 0.54–4.94; *p* = 0.006). After adjusting for other variables, the risk of lung cancer was lower and not significantly associated with an increase in the office and household workplace when compared to other occupations (Table 4).

## 4. Discussion

In this study, tobacco smoking and passive smoke exposure were associated with an increased risk of lung cancer. Alcohol consumption, history of lung disease and occupations with exposure to soot, iron and steel were also associated with having lung cancer.

Exposure to tobacco smoke is a well-established risk factor for lung cancer [7,19], and, in this study, exposure to both tobacco smoking and passive smoke at home or work were associated with a significantly increased risk of lung cancer. Lung cancer risk is estimated to be 20-fold for people that smoke cigarettes, and this risk increases with the duration and quantity consumed [7]. In this study we did not quantify smoking or characterize it in detail, but measurement by reporting use for tobacco was adequate. Exposure to passive smoke is also a well-known risk factor for lung cancer, which has been shown in the past few decades [20].

Tar found in cigarettes has approximately 3500 different compounds in it, and approximately 60–70 of the chemicals found in cigarettes are classified as carcinogens, 1,3-butadiene, ethyl carbonate and arsenic, just to name a few [19,21]. Some of the compounds in tobacco are either carcinogenic or they need to be metabolically activated [19,21]. These compounds are oxygenated by cytochrome P450 enzymes, which can result in DNA adducts that start a process of carcinogenesis [22,23]. All the different DNA adducts that are not repaired by the DNA repair mechanism can lead to the development of lung cancer [21,23].

Epidemiological studies have reported an inconsistent association between the risk of lung cancer and alcohol [7], and this study found that alcohol was a significant positive determinant of lung cancer. According to a review by Bandera et al., two case-control studies and three cohort studies found a significant dose-response association of total alcohol intake and lung cancer [24]. The risk estimates of the association ranged from the relative risk (RR) of 1.6 to 2.2 for more than 41 to 176 mL of drinks per week of pure ethanol per day (approximately 3000 mL of beer per day) [24,25,26]. Other studies have found that even consuming one glass of alcohol per day has the potential to increase the risk of lung cancer in both males and females (RR-1.23, 95%CI, 1.06–1.41). The mechanism of alcohol and lung cancer is not well established in humans; however, the studies on animals showed that ethanol increases the risk of lung cancer through the primary oxidative metabolism mechanism [24,27]. Rats that were used in experiments and those exposed to alcohol showed DNA damage [27]. Another study conducted by Fang and Vaca found that acetaldehyde, a first metabolite of ethanol oxidation, is mutagenic and carcinogenic in animal experiments, therefore compounding the risk by increasing DNA adducts in peripheral blood cells among alcoholics [28].

However, several studies indicate that lung cancer can also develop in never smokers [29,30,31] which means there are other risk factors that contribute to lung cancer. Lo et al. listed exposure to radon, passive smoke, occupational carcinogens and history of lung disease as some of the risk factors for lung cancer in patients that have never smoked [31]. Some of these exposures were also found to be positively associated with lung cancer in this study. This study found a higher lung cancer incidence in males than females, which is consistent with literature [2]. The most commonly and well-established risk could be that more men than women smoke tobacco and are exposed to occupational carcinogens [7,19]. Literature indicates that women are more likely to be exposed to the cooking fumes and indoor air pollution [32].

History of lung disease was found to be associated with lung cancer, and this is in line with other studies [14,33]. According to these studies, the people with a history or pre-existing lung disease such as tuberculosis (TB) and asthma are at a higher risk for lung cancer [33]. Young et al. reported a 5-fold (OR) increased risk for people with a history of lung disease and smoking compared to people who only smoke and do not have a history of lung disease [14]. The risk of TB is postulated to be through biological mechanisms where prolonged pulmonary inflammation leads to tissue damage, fibrosis, scars forming in the lungs and genomic damages [34]. Due to this inflammation and scarring that is associated with TB, it is possible that TB plays a role as either the initiator or promoter of lung cancer [35]. Another alternate explanation of this association of lung cancer with TB is the misdiagnosis of lung cancer for TB because of similar pulmonary symptoms and presentation [36]. Studies indicate that those with a lung cancer diagnosis are first diagnosed as TB before lung cancer [37,38] with one study having an incidence of 14.6% of patients with misdiagnosis of lung cancer as TB [36]. The misdiagnosis delays the cancer diagnosis therefore influencing prognosis.

This study found that occupations such as manufacturing and agriculture increased the risk of lung cancer which is a finding consistent with the literature [39,40]. We found that occupations that result in exposure to particulates and fumes such as in mining, construction and iron and steel increased the risk of lung cancer compared to people not exposed. For individuals that work in mines, the risk of lung cancer is increased by the exposure to inorganic metals in the form of fumes, alloy and hexavalent compounds [7,41] (4,29). Recent epidemiological evidence indicates a higher and increased lung cancer risk among stainless steel welders compared to mild steel welding workers [39]. This could be caused by the different quantity in toxicity of the welding fumes [42]. Men that were employed at baseline as welders were associated with a 16% increased risk of lung cancer. The risk was greater for welders in the vehicle repair industry at 40% greater than non-welders. These employees were exposed to carcinogenic exposures such as iron-containing steel, chromium, nickel and aluminum [39].

A positive association of lung cancer with exposure to asbestos and radon was found in this study; however, the association was not statistically significant. The literature shows asbestos as a widely known substance associated with lung cancer in occupational settings [43]. Individuals that work in construction are exposed to a mixture of risk factors such as asbestos and other fibers, painting and chemicals. Asbestos is used in construction building, and the association of asbestos and lung cancer is well established [7]. Painting has been associated with 40% excess risk of lung cancer; this is due to painters being exposed to hydrocarbon and chlorinated solvents, dyes, polyesters, phenol-formaldehyde and polyurethane resins [7]. Lung cancer is a disease, and the exposures associated with it may have complex interactions within the host.

The strength of this study is that exposure data to risk factors were collected directly from the participants and not through the medical records, therefore allowing a better characterization of exposure. Secondly, the cases and controls were selected from the same health facilities which reduced selection bias because participants are more likely to come from similar communities with a comparable socioeconomic status. Limitations in our study were that the sample size was likely to have been influenced by the high mortality rate of lung cancer because of late presentation to the health facilities, therefore missing a group of patients such as those who are living far from these treatment centers. The number of cases that had a histologic classification was limited due to missing data on the patients’ medical records and the basis of diagnosis used for the lung cancer patients. Further investigation on data collection needs to be conducted in the future to ensure that all the clinical data are recorded in the patient file.

## 5. Conclusions

This study provides insight into the epidemiological risk factors of lung cancer in the population of KwaZulu-Natal. The study suggests that although tobacco smoking and passive smoke exposure are major risk factors of lung cancer, increased exposure to some occupational and environmental substances may increase the risk of developing lung cancer. Alcohol consumption and history of lung disease are also important risk factors for lung cancer. The control of tobacco smoking, exposure to passive smoke and carcinogens are important for lung cancer prevention. Therefore, there is a need for the strengthening of the tobacco production policy and occupational health and safety policy, and for the expanding of lung cancer prevention and screening programs.

## Figures and Tables

**Figure 1 ijerph-19-06752-f001:**
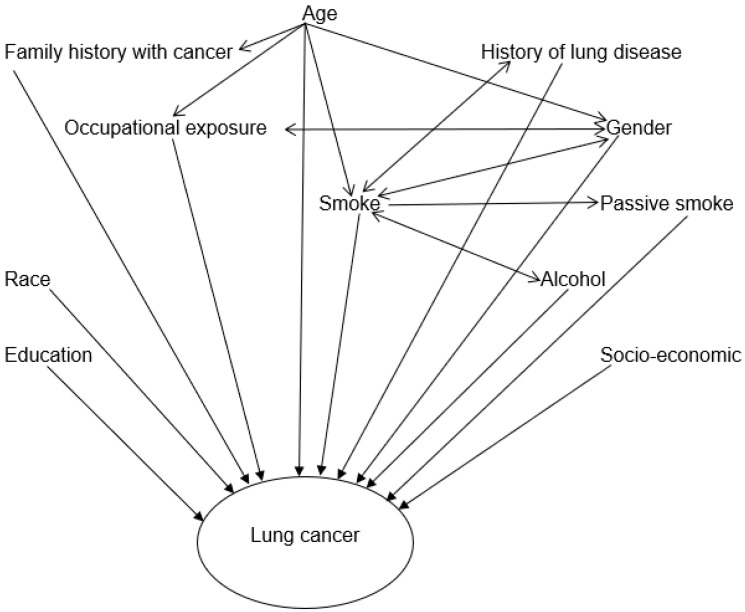
Directed acyclic graph illustrating the relationship among potential risk factors and lung cancer collected in the study.

**Figure 2 ijerph-19-06752-f002:**
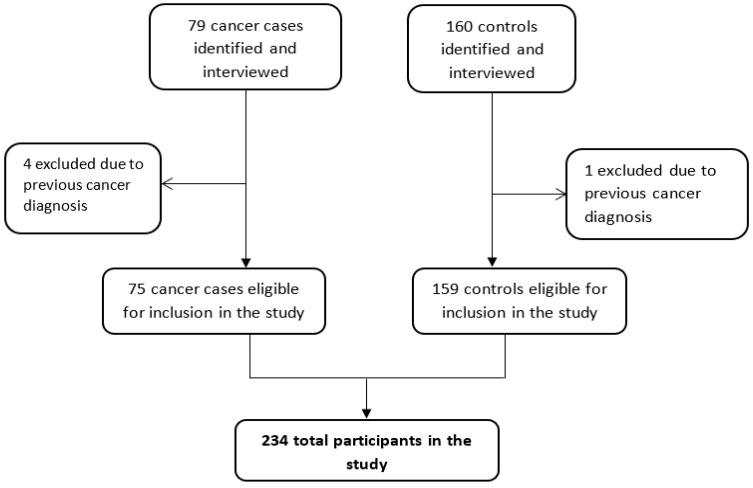
Flow diagram showing recruitment, inclusion and exclusion of participants for the study.

**Table 1 ijerph-19-06752-t001:** Frequency of general and clinical characteristics of lung cancer cases and controls participating in the study (*n* = 234).

	Characteristics/Category	Cases N (%)	Controls N (%)	Crude OR (95% CI)	*p*-Value
**Total no**		75	159		
**Age**					
	Mean ± SD (range)	±61.8 (13.9)	±54.8 (14.4)		
	21–39	7 (9.3)	16 (10.1)	-	
	40–49	8 (10.7)	48 (30.2)	0.38 (0.12–1.22)	**<0.001**
	50–59	12 (16.0)	40 (25.2)	0.69 (0.23–2.06)	
	60–69	25 (33.3)	32 (20.1)	1.79 (0.64–5.01)	
	70+	23 (30.7)	23 (14.5)	2.29 (0.79–6.60)	
**Gender**					
	Female	27(36.0)	63(39.6)	-	
	Male	48 (64.0)	96 (60.4)	1.17 (0.66–2.06)	0.595
**Race**					
	African	43 (57.3)	135 (84.9)	-	
	Mixed race	0 (0)	2 (1.26)	-	
	White	10 (13.3)	9 (5.7)	3.49 (1.33–9.14))	**<0.001**
	Asian	22 (29.3)	13 (8.2)	5.31 (2.47–11.4)	
**^a^ Marital status**					
	Single	17 (23.3)	69 (44.5)	-	
	Married	42 (57.5)	67 (43.2)	2.54 (1.32–4.90)	**0.004**
	Divorced + Widowed	14 (19.2)	19 (12.3)	2.99 (1.25–7.14)	
**^b^ Education**					
	No education—Primary	20 (30.3)	40 (29.0)	-	
	High school—Higher education	46 (69.7)	98 (71.0)	0.94 (0.49–1.78)	0.847
**^c^ Monthly household income**	0–R4000	44 (71.0)	77 (74.0)	-	
	R4 001+	18 (29.0)	27 (26.0)	1.17 (0.58–2.35)	0.667
**Clinical characteristics of cases**					
**Basis of diagnosis**					
	Biopsy	56 (74.7)	-		
	x-ray	10 (13.4)	-	-	-
	CT Scan	6 (8.00)	-		
	Cytology	3 (4.00)			
	^d^ Histological classification			-	-
	Adenocarcinoma	16 (21.3)	-		
	Squamous Cell Carcinoma	11 (14.7)	-		
	Other	9 (12)	-		
**Staging**					
	1	1 (1.33)	-		
	2	0 (0.00)	-	-	-
	3	2 (2.67)	-		
	4	28 (37.3)	-		
	Unknown	44 (58.7)	-		

Some variables had “unknown and or participants refused to answer”; therefore, data analysis was conducted on incomplete dataset of those variables; In bold: *p*-value of ≤ 0.05, a: *n* = 228; b: *n* = 204; c: *n* = 166; d: *n* = 36.

**Table 2 ijerph-19-06752-t002:** Frequency of occurrence of lifestyle, occupational and environmental risk factors of lung cancer amongst the cases and controls (*n* = 234).

Variables		Cases *n* = 75 (%)	Controls *n* = 159 (%)	OR (95% CI)	*p*-Value
**Lifestyle**					
Passive smoke exposure	No	34 (45.3)	112 (70.4)	-	
	Yes	41 (54.7)	47 (29.6)	**2.87 (1.63–5.07)**	**<0.001**
Have you ever smoked	No	26 (34.7)	102 (64.2)	-	
	Yes	49 (65.3)	57 (35.9)	**3.37 (1.90–6.00)**	**<0.001**
Have you ever consumed alcohol?	No	50 (66.7)	93 (58.5)	Ref	
	Yes	25 (33.3)	66 (41.5)	**2.82 (1.59–5.00)**	**<0.001**
**Family history with cancer**					
Biological father	No	73 (97.3)	158 (99.4)	-	
	Yes	2 (2.67)	1 (0.63)	4.33 (0.39–48.50)	0.235
Biological mother	No	69 (92.0)	155 (97.5)	-	
	Yes	6 (8.00)	4 (2.52)	3.37 (0.92–12.30)	0.066
Siblings	No	64 (85.3)	156 (98.1)	-	
	Yes	11 (14.7)	3 (1.89)	**8.93 (2.41–33.10)**	**0.001**
* History of lung disease	No	53 (70.7)	151 (95.0)	-	
	Yes	22 (29.3)	8 (5.03)	**7.83 (3.29–18.7)**	**<0.001**
**Employment sector**					
Office and Household	No	29 (38.7)	51 (32.1)	-	
	Yes	46 (61.3)	108 (67.9)	0.75 (0.42–1.33)	(0.322)
Transport, Storage and Repair of motor vehicles	No	59 (78.7)	146 (91.8)	-	
	Yes	16 (21.3)	13 (8.18)	**3.05 (1.38–6.72)**	**(0.006)**
* Mining, Construction and Manufacturing	No	40 (53.3)	113 (71.1)	-	
	Yes	35 (46.7)	46 (28.9)	**2.15 (1.22–3.79)**	**(0.008)**
Agriculture, Forestry and Fishing	No	65 (86.7)	151 (95.0)	-	
	Yes	10 (13.3)	8 (5.03)	**2.90 (1.09–7.69)**	**(0.032)**
**Occupational exposures**					
Soot	No	68 (90.7)	157 (98.7)	-	
	Yes	7 (9.33)	2 (1.26)	**8.08 (1.64–39.9)**	**(0.010)**
Iron and Steel	No	67 (89.3)	157 (98.7)	-	
	Yes	8 (10.7)	2 (1.26)	**9.37 (1.94–45.3)**	**(0.005)**

*p*-value of ≤ 0.05, Significant OR and 95 % CI were marked in bold style; * Mining was not classified according to minerals mined, e.g., coal, asbestos, etc.

**Table 3 ijerph-19-06752-t003:** Associated risk factors in different logistic models for tobacco smoke and passive smoke exposure.

Variable		Univariate OR (95%CI)	Multivariate Models aOR (95% CI)	
			Tobacco Smoking	Passive Smoke Exposure
Tobacco smoking	No			-
	Yes	*** 3.37 (1.90–6.00)** **(<0.001)**	**** 2.86 (1.21–6.77)** **(0.017)**	-
Passive smoke exposure	No		-	
	Yes	*** 2.87 (1.63–5.07)** **(<0.001)**	-	**** 3.28 (1.48–7.30)** **(0.004)**
Alcohol consumption	No			
	Yes	*** 2.82 (1.59–5.01)** **(<0.001)**	**** 2.79 (1.21–6.41)** **(0.016)**	**** 3.35 (1.44–7.75)** **(0.005)**
Sibling with cancer	No			
	Yes	**** 8.94 (2.41–33.1)** **(0.001)**	3.83 (0.53–27.4) (0.181)	3.79 (0.49–29.2) (0.201)
* History of lung disease	No			
	Yes	*** 7.83 (3.29–18.7)** **(<0.001)**	*** 9.91 (3.04–32.3)** **(<0.001)**	*** 12.1 (3.72–39.7)** **(<0.001)**

All multivariate models were adjusted for age, gender, race, marital status, education, alcohol consumption, sibling and history of lung cancer disease; OR: odds ratio; aOR: adjusted odds ratio; * History of lung disease (tuberculosis, asthma, pneumonia or chronic bronchitis); ** *p*-value < 0.001; * *p*-value < 0.05; *p*-value of ≤ 0.05, Significant OR& aOR and 95 % CI were marked in bold style.

**Table 4 ijerph-19-06752-t004:** Crude and adjusted odds ratio of different logistic models of lung cancer associated with different occupational risk factors.

Variables		Univariate OR (95% CI)	Multivariate aOR (95%CI)			
			Office and Household	Transport, Storage and Repair of Motor Vehicles	Mining, Construction and Manufacturing	Agriculture, Forestry and Fishing
Office and Household	No			-	-	-
	Yes	0.75 (0.42–1.33) (0.322)	0.69 (0.30–1.62) (0.395)	-	-	-
Transport, Storage and repair of motor vehicles	No		-		-	-
	Yes	*** 3.05 (1.38–6.72)** **(0.006)**	-	1.63 (0.54–4.94) (0.387)	-	-
Mining, Construction and Manufacturing	No		-	-		-
	Yes	*** 2.15 (1.22–3.80)** **(0.008)**	-	-	*** 2.64 (1.16–6.01)** **(0.020)**	-
Agriculture, forestry and fishing	No		-	-		
	Yes	*** 2.90 (1.10–7.69)** **(0.032)**	-	-		3.69 (0.95–14.3) (0.059)
Alcohol consumption	No					
	Yes	**** 2.82 (1.59–5.01)** **(<0.001)**	*** 2.85 (1.23–6.60)** **(0.015)**	*** 2.72 (1.18–6.29)** **(0.019)**	*** 2.81 (1.19–6.60)** **(0.018)**	*** 2.76 (1.19–6.43)**
Sibling with cancer	No					
	Yes	*** 8.94 (2.41–33.1)** **(0.001)**	3.55 (0.51–24.8) (0.202)	3.30 (0.44–24.9) (0.246)	4.25 (0.61–29.6) (0.144)	4.23 (0.59–30.1) (0.150)
^a^ History of Lung disease	No					
	Yes	*** 7.83 (3.29–18.7)** **(<0.001)**	*** 10.6 (3.20–35.2)** **(<0.001)**	*** 9.63 (2.98–31.1)** **(<0.001)**	*** 9.73 (2.96–32.0)** **(<0.001)**	*** 9.98 (3.02–33.0)** **(<0.001)**
Tobacco smoking	No					
	Yes	*** 3.37 (1.90–6.00)** **(<0.001)**	**** 2.83 (1.19–6.70)** **(0.018)**	**** 2.86 (1.20–6.82)** **(0.017)**	**** 2.90 (1.20–7.00)** **(0.018)**	**** 2.86 (1.19–6.83)** **(0.018)**

All multivariate models were adjusted for age, gender, race, marital status, education, alcohol consumption, sibling, history of lung disease and smoking of tobacco; OR: odds ratio; aOR: adjusted odds ratio; ^a^ History of lung disease (tuberculosis, asthma, pneumonia or chronic bronchitis); ** *p*-value < 0.001; * *p*-value < 0.05; *p*-value of ≤ 0.05, Significant OR& aOR and 95 % CI were marked in bold style.

## Data Availability

Data from this study are the property of the KwaZulu-Natal Department of Health (KZN DOH) and the University of KwaZulu-Natal and cannot be made publicly available. All interested readers can access the data set from the DOH Research ethics committee and the University of KwaZulu-Natal Biomedical Research Ethics Committee (BREC) from the following contacts: The Health Research and Knowledge Management, 330 Langalibalele Street, Private bag X9051, Pietermaritzburg, 3200, Tel.: +27-33-3952805 Fax: +27-33-3943782 Email: hrk@kznhealth.gov.za The Chairperson Biomedical Research Ethics Administration Research Office, Westville Campus, Govan Mbeki Building University of KwaZulu-Natal P/Bag X54001, Durban, 4000 KwaZulu-Natal, South Africa Tel.: + 27-31-2604769 Fax: + 27-31-2604609 Email: BREC@ukzn.ac.za.

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
