# Peer review of "Epidemiological Study of Risk Factors for Lung Cancer in KwaZulu-Natal, South Africa"

_ijerph, 2022, doi:10.3390/ijerph19116752_

Round 1
Reviewer 1 Report
I appreciate that the authors have addressed the comments. The revised version of the manuscript “Epidemiological study of risk factors for lung cancer in Kwa-2 Zulu-Natal, South Africa” has certainly improved.
The authors should revise the text carefully before accepting.
Author Response
The authors appreciate the reviewer’s comment. The author has revised the text in the manuscript..

Reviewer 2 Report
The addition of a few details in the introduction is great; the clear prevalence of lung cancer globally and in South Africa and the differences in cancer prevalence by gender is a great addition to the manuscript. The addition of the DAG not only highlights all the variables included in the study but also shows the relationship of those variables with lung cancer; a great addition to the manuscript. A few minor editions;
Line 56-58: The prevalence of smoking of tobacco smoking prevalent has increased to 37% for males and significantly dropped to 8% for 57 women aged 15 and older[17] .
Line 58-59: The South African demographic and health survey indicated that among smokers, most smokers’ smoke cigarette on a daily basis at 30% for males and 6% for females smoke cigarettes on a daily basis [17].
Linen70-71: Cases were patients with the lung cancer diagnosis that was laboratory confirmed through a biopsy, X-ray, CT Scan and cytology, and or clinical investigations. This sentence still makes it sound like you had access to their medical files. It is clear that they obviously had cancer since they were being treated at a cancer clinic/hospital but I would caution wording it differently or stating that all details regarding cancer diagnoses, stage, comorbidities, and other exposure were self-reported during patient interviews right after the above sentence.
Author Response
- The author has taken the reviewer’s suggestion. The sentence has been edited as recommended on line 56-58.
- The author has taken the reviewer’s suggestion. The sentence has been revised as recommended on line 58-59.
- The author appreciates the reviewer’s comment. The author has revised the statement as suggested on line 73.

Reviewer 3 Report
NA
Author Response
Thank you for accepting our manuscript.
This manuscript is a resubmission of an earlier submission. The following is a list of the peer review reports and author responses from that submission.
Round 1
Reviewer 1 Report
This is a nice study but it doesn’t add anything to the current knowledge of lung cancer causes. Tobacco is the main cause of lung cancer.
Author Response
The authors appreciate the reviewer’s comment. However, there are studies that have been conducted on ‘non-smokers’ that have indicated that tobacco is not the main cause of lung cancer. This study was aimed at identifying a range possible risk factors in the province of KwaZulu-Natal.
Reviewer 2 Report
Epidemiological study of risk factors for lung cancer in Kwa-Zulu-Natal, South Africa
Considering the impact of mining and construction on cancer in the predominantly mining region of the country, this study of lung cancer in Kwa-Zulu Natal is a very interesting topic. The overall consideration of environmental, occupational, and biological factors and their effect on cancer incidence is important and very well thought out in this manuscript. However, I think this paper could be improved by providing a better definition for identification and selection of cases and controls, clearly defining and showing a categorization scheme for covariates included and expanding a little bit on the strengths and limitations of the study. My specific comments are below;
Background
Line 32-36; are these global statistics? If they are, make sure you mention that. Also provide South Africa specific lung cancer prevalence and incidence numbers and add a quick summary of studies of lung cancer in the country.
Line 48-49: Is the smoking prevalence statistics from 1998 to 2003 or is it statistics for the two individual years? The sentence is a little confusing to me.
Line 49: consider re-writing the sentence, e.g. “63.4% in men and 50.2% in women. However, in 2012 the estimated rate of tobacco smoking…”
Identification and recruitment of participants
Line 60-64: You mention that cases were recruited from the oncology departments of 3 health facilities. How did you recruit cases? Did you recruit them at time of clinical visit, get their consent then access their medical records to ensure diagnoses confirmation through biopsy and cytology? Or was the confirmation of diagnoses based on their response to the questionnaire? The chronological order of recruitment is not very clear here. If cases were based on questionnaire responses, I would be very careful with the language here since you did not actually confirm diagnoses but based diagnoses on patient’s response to specific questions.
Line 66-69: for your controls, did you do patient file reviews to select the controls? Did you enroll individuals first then do patient file review or was this based on questions you ask the patients at recruitment? If you asked them a specific question, I would include it here. Again, since this is based on responses to a questionnaire, I would suggest being very clear with the language used and also providing examples of key questions asked to determine controls versus cases.
In general, I would recommend including the informed consent statement (line 289-290) in the “Identification and recruitment of participants” section.
Line 85 - 86: how was passive smoke and household conditions defined?
Data analysis:
How was “history of lung disease” categorized? This is only evident in the results that it includes tuberculosis and asthma (line 139-140). Also there are no details on how demographic variables were categorized; the only time we see how variables are grouped is in the results table grouping of lifestyle, family history with cancer, and occupational exposures. I would also caution against grouping African and mixed race into one group since their socioeconomic factors tend to vary greatly and their environmental and occupational exposures are also very different.
Line 97: The occupations or industries that the participants were working in or have worked in
Results:
Line 130-131: The mean (SD) age was a higher for the cases 61.8 (13.9) years than for the controls 54.8 (14.4) years.
Table 1; was the information missing or unknown for a-c? Since this information is based on a questionnaire, I would assume participants did not know the answer or refused to respond. I would just clarify here which is which. If information is missing; does this mean the question was not asked?
Line 143: you mention that “More than half (58.7%, n=44) of the cancer cases were not staged”. In table 1, this was unknown staging based on questionnaire, are you assuming that if the patient did not know if the cancer was staged or not then it was not staged? I would recommend rephrasing this to say ‘the stage of lung cancer was unknown for More than half (58.7%, n=44) of the cancer cases.’
Line 166-167: a n=228 | b: n204 , there is no assignment for a or b in the table
Line 180-181: is it history of lung disease or history of lung cancer? I am assuming you meant lung disease since individuals with history of lung cancer were excluded from the study.
Discussion
Great discussion about the effect of smoking, alcohol, mining &construction exposures. I would recommend adding a little bit about the differences in lung cancer by gender in your study; More males than females with lung cancer. Is this due to male specific exposures such as work or smoking status? How is smoking different for males and females in South Africa?
Line 259-260: I would not refer to the use of personnel interviews/questionnaire without medical chart review/verification for diagnoses a strength of the study. This seems like the biggest limitation to me; you are relying on individual’s answers to questions without the lab/biopsy/cytology verification in medical records to confirm diagnoses. This introduces so many issues with bias. If the questionnaire was in addition to medical records review or some other form of confirmed registry for cancer, then this would be considered a strength.
Author Response
Background
- Yes, these are the global statistics. The author has edited the sentence on Line 42.
The author has also included the South African lung cancer stats on Line 34 - 35. - The author has eliminated this sentence. The author added updated smoking prevalence statistics on Line 56 – 60.
- The author has taken the reviewer’s suggestion and re-wrote the sentence on line55-58.
Identification and recruitment of participants
- The author has amended the statement to make it clear and added that ‘The cases were recruited at the oncology clinic during the patient’s scheduled lung cancer treatment appointment at the three public health facilities.’.’ Line 71 -73.
- The author has noted the reviewer’s comment and added information for clarity on the recruitment of controls on Line 76 - 82
- The author appreciated the reviewer’s recommendation. A blank informed consent was sent to the journal as an attachment.
- The author has noted the reviewer’s comment and included how passive smoke was defined in the study on Line 121 - 122.
The household conditions were eliminated.
Data analysis:
- The author has taken the reviewer’s comment and added information on how the ‘history of lung cancer’ and demographic variables were catergorised on Line 119 – 124.
The author has also separated the African and mixed race in Table 1 - The author has taken the reviewer’s suggestion and re-wrote the sentence on line 124 - 125.
Results:
- The author has taken the reviewer’s suggestion and re-wrote the sentence on line 158- 159
- The author has noted the reviewer’s comment and added information to clarify that the information is not missing but the participants didn’t know the answer or refused to respond. Line –166.
- The author appreciates the reviewer’s suggestions, the sentence has been rephrased on Line 169 – 170.
- The author has eliminated ‘a’ and ‘b’.
- The author has amended accordingly to history of lung disease. Line 212
- The author has taken the reviewer’s suggestion and added information on lung cancer by gender on line 254– - 258
- The author has noted the reviewer’s comment. The diagnosis was confirmed via medical records and hospital-based cancer registry.
Reviewer 3 Report
P.Mbeje et al. studied risk factors associated with lung cancer using a case control study in KwaZulu Natal, South Africa. They found tobacco smoking and passive exposure to be associated with increased risk of lung cancer. The other factors were alcohol consumption, lung disease and exposure to environmental chemicals. The authors address an important public health question, as identifying at-risk group is also relevant since it will direct intervention strategies. Yet, I have a few major concerns (described below), which I hope the authors could address.
Major comments
- The entire manuscript needs thorough English editing. There are several places where the sentence formation is not complete and has grammatical errors.
- Can the authors confirm that they followed the STROBE guidelines both while writing the manuscript and planning the study?
- I suggest including the directed acyclic graph of the study question which will present the main effects, potential confounders and the role of effect modifiers.
- Because lung cancer is known to have a long latency, and because factors like air pollution, cigarette smoking, and occupational exposure are being investigated in relation to lung cancer deaths. The authors should mention the possibility of having complex interactive effects to reach meaningful conclusions
- Table 3 and 4 : I think the results can be presented in clear away. I am not sure if you need the “Ref” mentioned in that format.
- Literature clearly shows that major differences exist between males and females with respect to risk factors for lung cancer. In males, cigarette smoking and occupational exposure are important considerations, whereas in females, indoor air pollution appears to be the most significant factor. This observed differences should be mentioned considering the % of males and females in the study.
Author Response
- The author confirms to using the STROBE guidelines.
- The author has included the direct acyclic graph (Figure 1)
The author appreciates the reviewer’s suggestion and has added information on line 295 – 296. - The author has taken the reviewer’s suggestion and amended Table 3 and 4.
- The author appreciates the reviewer’s suggestion and has added information on lung cancer by gender on line 254 - 258.
Reviewer 4 Report
This manuscript explores the potential risk factors associated with lung cancer among patients seen in the public health facilities in KwaZulu-Natal, South Africa with a statistical method. The author found tobacco smoke exposure is a major cause of lung cancer, and increased exposure to occupational and environmental carcinogenic substances, alcohol consumption and history of lung disease is also increasing the risk of lung cancer. Although the conclusion of this manuscript is well-known, this work still can provide a reference for the formulation of lung cancer prevention and control policies in Africa.
The topic is interesting, and the manuscript is well-organized and well-written. However, the following major revisions need to be considered before consideration for publication.
Specific comments:
- In the Introduction section, “In South Africa, 47 some provinces in 1998 and 2003 used to have a prevalence of tobacco smoke of up-to 48 63.4% in men and 50.2% in women however, in 2012 the estimated rate of tobacco smoking 49 dropped to 33% and 31% respectively”. Readers may be more concerned with the detailed investigation of the whole of South Africa in recent years. In addition, there is any national epidemiology of lung cancer and smoking? The author should reference several works of literature and/or reports to address the relationship between smoking and lung cancer rate.
- In the Materials and Methods section, the authors should give more information on the statistical analysis, including the detailed analysis method and software used in this study.
- There are various forms of smoking, such as cigarette smoking, secondhand smoking, and E-cigarettes. The author should give a clear explanation.
- In Table 1, “Basis of diagnosis” section, What’s the “other” represent? It seems that only 75% of patients have clear diagnostic criteria, how about the other 25% of patients? In the “Histological classification” section, only 36% of patients are diagnosed with lung cancer. What’s the “other” represent? Does this mean that in most cases, it is impossible to distinguish between lung cancer and lung metastatic tumor?
- Whether the risk of lung cancer would rise with the degree of smoking. The author divided the samples into two groups according to whether smoking or not.
- References should be uniform according to the journal format.
Authors Response
- The author has taken the reviewer’s suggestion and added information on Line 34 – 40 and Line56 – 60.
- The author has taken the reviewer’s suggestion and added information on Line 119 – 1248.
The author used REDCap database and Stat IC version 13 for analysis. Line 117 - 118 - The author has noted the reviewer’s comment and has clarified that it is tobacco smoke (cigarette) on Line 98 and 121
- The author has noted the reviewer’s comment. The ‘other’ was eliminated and the different types of diagnosis was written. All patients were diagnosed with lung cancer. However, some cases did not have histological classification.
The author has clarified this on the Table 1 and on the limitations on Line 304 – 308. - All patients were diagnosed with lung cancer. However, some cases did not have histological classification.
The author has clarified this on the Table 1 and on the limitations on Line 304 – 308. - The author appreciates the reviewer’s comment and has amended accordingly.
Reviewer 5 Report
- Add more reference from latest published work to support this study in introduction section.
- 22% plagiarised, reduce up to acceptable range.
Author Response
The author has taken the reviewer’s comments and has amended accordingly.